# Emerging Blood-Based Biomarkers for Predicting Immunotherapy Response in NSCLC

**DOI:** 10.3390/cancers14112626

**Published:** 2022-05-26

**Authors:** Ana Oitabén, Pablo Fonseca, María J. Villanueva, Carme García-Benito, Aida López-López, Alberto Garrido-Fernández, Clara González-Ojea, Laura Juaneda-Magdalena, Martín E. Lázaro, Mónica Martínez-Fernández

**Affiliations:** 1Translational Oncology Research Group, Galicia Sur Health Research Institute (IIS Galicia Sur), Sergas-Uvigo, 36204 Vigo, Spain; anaoitaben@gmail.com (A.O.); fonsecasilvapab@gmail.com (P.F.); maria.jose.villanueva.silva@sergas.es (M.J.V.); carme.garcia.benito@sergas.es (C.G.-B.); aida.lopez.lopez@sergas.es (A.L.-L.); alberto.garrido.fernandez2@sergas.es (A.G.-F.); clara.gonzalez.ojea@sergas.es (C.G.-O.); laura.juaneda.magdalena.benavides@sergas.es (L.J.-M.); martin.emilio.lazaro.quintela@sergas.es (M.E.L.); 2Genomes and Disease Lab, CiMUS, Av. Barcelona, Universidade de Santiago de Compostela, 15706 Santiago de Compostela, Spain; 3Department of Zoology, Genetics and Physical Anthropology, Universidade de Santiago de Compostela, 15706 Santiago de Compostela, Spain; 4Posada Lab, CINBIO, Universidade de Vigo, 36310 Vigo, Spain; 5Medical Oncology Department, Hospital Álvaro Cunqueiro, Sergas, 36213 Vigo, Spain; 6Pharmacy Department, Hospital Álvaro Cunqueiro, Sergas, 36213 Vigo, Spain; 7Anatomical Pathology Department, Hospital Álvaro Cunqueiro, Sergas, 36213 Vigo, Spain; 8Galicia Sur Health Research Institute (IIS Galicia Sur), Hospital Álvaro Cunqueiro, Carretera Clara Campoamor No. 341, 36213 Vigo, Spain

**Keywords:** soluble biomarkers, immunotherapy, ICI, NSCLC, PD-L1, ncRNA, cytokines, ctDNA, bTMB

## Abstract

**Simple Summary:**

Treatment with immunotherapy has been established as a standard treatment for lung cancer in recent years. Unfortunately, still, only a small proportion of patients benefit from the treatment, being the first leading cause of cancer death worldwide. Therefore, there is an urgent need for predictive biomarkers to help clinicians to discern whose patients are more likely to respond to immunotherapy. Since liquid biopsy opens the door to select patients and monitor the response during the treatment in a non-invasive way, in this review, we focus on the most relevant and recent results based on blood soluble biomarkers.

**Abstract:**

Immunotherapy with Immune Checkpoint Inhibitors (ICIs) has demonstrated a profitable performance for Non-Small Cell Lung Cancer (NSCLC) cancer treatment in some patients; however, there is still a percentage of patients in whom immunotherapy does not provide the desired results regarding beneficial outcomes. Therefore, obtaining predictive biomarkers for ICI response will improve the treatment management in clinical practice. In this sense, liquid biopsy appears as a promising method to obtain samples in a minimally invasive and non-biased way. In spite of its evident potential, the use of these circulating biomarkers is still very limited in the real clinical practice, mainly due to the huge heterogeneity among the techniques, the lack of consensus, and the limited number of patients included in these previous studies. In this work, we review the pros and cons of the different proposed biomarkers, such as soluble PD-L1, circulating non-coding RNA, circulating immune cells, peripheral blood cytokines, and ctDNA, obtained from liquid biopsy to predict response to ICI treatment at baseline and to monitor changes in tumor and tumor microenvironment during the course of the treatment in NSCLC patients.

## 1. Introduction

Although solid-tumor cell detection in blood has long been established for prognostic purposes, with the discovery of shed tumor molecules and cells in bodily fluids dating back more than 100 years ago [1], it was not until as recently as 2010 that the term “liquid biopsy”, referring to circulating solid-tumor cells in the blood (as opposed to tissue) was coined [2]. Curiously enough, in the same year, the TNM AJCC 7th edition included the designation cM0(i+) to describe, among other scenarios, the presence of circulating tumor cells in peripheral blood when no clinical or radiological evidence of metastasis is found.

Since then, the ability to study non-hematologic cancers through blood or other bodily fluids samples has taken the oncology community by storm, with hundreds of thousands of publications in the last decade, studying not only tumor cells but also tumor molecules, such as circulating tumor DNA (ctDNA), circulating non-coding RNAs (circ ncRNAs), proteins, cytokines, tumor organelles including exosomes, as well as immune cells and tumor-educated platelets, neutrophils, eosinophils, becoming definitely one of the most rapidly evolving fields in the field of cancer [3,4,5]. Circulating tumor cells (CTCs, cells that enter the bloodstream from the primary or metastatic tumors) have been broadly studied as circulating biomarkers for cancer management. However, the low content of CTCs in the peripheral blood, together with their heterogeneity (expressing different markers dependent on the CTC population), makes their isolation and enrichment a hard challenge, consequently making their translation into the clinical routine really difficult.

Advances in liquid biopsy have been pioneered by targeted therapy in lung cancer, driven both by the need for accurate measurements of critical biomarkers for prognosis and treatment purposes as well as advances in detection techniques, not to mention the implementation of bioinformatics with the so-called supervised machine learning and the development of multidisciplinary molecular tumor boards for result deciphering and subsequent application in clinical practice.

The advantages of liquid biopsies over tissue biopsy have been broadly manifested: it is a minimally invasive method, it allows serial samples and monitoring of molecular changes throughout the course of the disease, and it avoids the loss of information due to tumor heterogeneity since the DNA is obtained from all of the tumor sites, as opposed to a tissue biopsy in which a sample is taken from a specific site, not to mention the difficulty of obtaining valid samples from some metastatic locations, a fact of special relevance in lung cancer [6]. Therefore, in the metastatic setting, not only does liquid biopsy enable the detection of different phenotypes within a single tumor and between metastatic sites (the named tumor spatial heterogeneity), but it can also capture tumor changes over time (tumor temporal heterogeneity), providing insights into the dynamic changes occurring in the tumor and its environment, whether spontaneous or drug-induced, with consequent potential therapeutic implications [6].

Liquid biopsy harbors great potential for clinical applications, such as early diagnosis, recurrence prediction with detection of minimal residual disease, prediction of tumor response/progression/resistance, and treatment selection guidance [6], most of which are yet to be perused, with only the predictive factor option being ready for clinical use in oncology and most specifically in lung cancer.

As far as lung cancer is concerned, it represents the second most common cancer in incidence, with the Non-Small Cell Cancer (NSCLC) subtype accounting for 84% of all lung cancer diagnoses [7]. Its therapeutic landscape has dramatically changed over the last decade because of both the advent of targeted therapy (with a rapidly growing number of driver mutations identified in recent years) and the dazzling irruption on the scene of Immune Checkpoint Inhibitors (ICI), a groundbreaking immunotherapy treatment with unprecedented long-term survival in those patients achieving a complete response: the so-called *tail of the curve*. Immunotherapy has become standard clinical practice either in combination with chemotherapy (in the first line, in patients with PD-L1 expression <50%) or as monotherapy (in the first line, provided a PD-L1 expression ≥50%, as well as in the second line), with a 5-year overall survival (OS) of 20% in unselected and up to 40–50% in PD-L1 high expressing patients [8,9,10].

Despite those improvements, NSCLC remains the leading cause of cancer death in the Western world [7], with a desperate need for biomarker identification to improve treatment outcomes, patient selection, and avoid unnecessary toxicity.

Since ICIs have been shown to have a large impact on a small proportion of patients, attempts have been made to improve patient selection and subsequent treatment outcomes. In brief, the interaction between tumors and the immune system consists of eliciting an immune response through neoantigen production, presented by the human leukocyte antigen (HLA) type II complex, located on the membrane of antigen-presenting cells (such as dendritic cells, macrophages, or B cells), to the T cell receptor (TCR) found on the membrane of the inactivated T lymphocyte, thus inducing the priming or activation phase of the lymphocyte. Once in the peripheral tissues, the activated lymphocyte searches for neoantigen-labeled tumor cells to eradicate them, the effector phase. Mission accomplished: lymphocytes have a programmed death receptor (PD-1) to undergo apoptosis. Importantly, tumors harbor the ligand to PD-1 (PD-L1) to trigger lymphocyte apoptosis and elude tumor death. Allegedly, the higher the PD-L1 count, the greater the likelihood of a response. Indeed, the percentage of PD-L1 expression in tumor tissue has been shown to predict response to ICIs as monotherapy, especially in the first-line setting of NSCLC (*p* < 0.001) [11,12]. This correlation between PD-1/PD-L1 expression and response rate, together with the drug-diagnostic co-development model, has prompted organizations such as the FDA and EMA to establish PD-1/PD-L1 assays as drugs companion diagnostics, meaning that the indication of these drugs rely on the result of the assays, which are mandatory in order for the drug to be used. Of note, high PD-L1 expression (≥50%) has been correlated with current/ever smoking as well as with a higher overall response rate (ORR) among current/former smokers [13]. However, some patients with low PD-1/PD-L1 expression have been proved to be good responders, probably due to intratumor heterogeneity in tissue biopsies and/or to other molecular mechanisms involved in ICI response that are currently missed. Tumor mutational burden (TMB), defined as the total number of somatic mutations per megabase detected in a tumor, has also been related to tumor immune response. Its utility lies in the fact that abnormal proteins are recognized as tumor neoantigens, which in turn elicit the activation of the immune system. Nevertheless, the use of this parameter as a predictive response biomarker in clinical practice has some important limitations, such as differences in the threshold used to define “high” TMB, again the bias caused by the intratumor heterogeneity, or the fact that patients showing low TMB become good responders to immunotherapy. Therefore, liquid biopsy becomes a promising tool providing a surrogate source of the overall tumor tissue status, reflecting a panoramic view of the whole molecular profile in real-time. Different markers such as ctDNA, circncRNAs, cytokines, angiogenic factors, and the immune cells themselves have all been implicated in tumor immune response. The current review aims to walk the reader through the latest evidence on those emerging predictive biomarkers for ICIs using liquid biopsy along with treatment, as shown in Figure 1.

## 2. Soluble and Exosomal PD-L1

Programmed death ligand 1 (PD-L1) is expressed on the surface of tumor cells but also on the surface of different immune cells as well as on other cells in the tumor microenvironment. In addition to being located on the cell surface, PD-L1 is also encountered in extracellular forms, such as exosomal PD-L1 (exoPD-L1) or free-form PD-L1 (soluble PD-L1, sPD-L1). All of these forms of PD-L1 expression have been shown to be able to inhibit anti-tumor immune responses [14].

Even though sPD-L1 is mostly derived from the membrane cleavage of tumor PD-L1 [15], other sources of PD-L1, such as myeloid cells and activated T cells, have high levels of membrane PD-L1, with myeloid cells being the immune cells with the greatest capacity to release sPD-L1 [16]. This suggests the presence of a distinct regulatory mechanism for PD-L1 release, different from PD-L1 production [16]. sPD-L1 is readily detectable in peripheral blood, and several studies suggest that it maintains its biological PD-1-binding capacity, exerting an immunosuppressive function. [17] In addition, variants of sPD-L1 have been found to contribute to therapeutic resistance, neutralizing anti-PD-L1 antibodies in a dose-dependent manner or forming covalently-linked homodimers with greater inhibitory effect than the monomeric forms [18]. The detection of those variants has been postulated to be more accurate predictive biomarkers than the sPD-L1 levels themselves [15].

Nonetheless, studies analyzing the role of sPD-L1 in different tumor types have so far produced inconsistent and even conflicting results. In a meta-analysis performed by Cheng Y et al., which included 1188 lung cancer patients, the presence of high pre-treatment sPD-L1 levels predicted worse overall survival (OS), progression-free survival (PFS), and lower ORR to both ICI and non-ICI agents, suggesting sPD-L1 as a potential predictive biomarker in lung cancer [16]. In general, elevated sPD-L1 levels appear to be associated with a lower response rate and worse prognosis in ICI treatment, with a significant impact on PFS and OS, as shown in a cohort of 128 patients with lung cancer (both small cell and non-small cell subtypes), melanoma and bladder cancer, in which an sPD-L1 level > 11 pg/μL (high sPD-L1) proved to be an independent prognostic factor for decreased PFS and OS, with differences of up to 3.4 and 6 months, respectively, supporting the fact that pre-treatment plasma sPD-L1 levels can be used to predict ICI response, PFS and OS in advanced solid tumors such as in melanoma, bladder, and lung cancer [19].

Regarding the variation of PD-L1s during treatment, So Yeon Oh et al. analyzed pre- and post-treatment sPD-L1 levels in 67 patients with NSCLC, melanoma, small cell lung cancer (SCLC), urothelial carcinoma, and other cancers. sPD-L1 levels increased during the first weeks but correlated differently depending on the so-called “immunogenic” tumor type: while in NSCLC patients, an increase of >100% in sPD-L1 levels during the first weeks of ICI treatment was significantly associated with better outcomes (both PFS and OS), in melanoma patients, the same increase in sPD-L1 levels after initiating treatment showed a reverse relationship with worse PFS outcomes (0.9 vs. 5.7 months) [19]. However, the true value in sPD-L1 level changes has not been established since the results in NCSLC are not consistent across the different studies reviewed [15,16,17,18,20].

Exosomes are small extracellular lipid bilayer vesicles measuring between 30 and 100 nm in diameter and being present in almost all body fluids. Secreted into the extracellular space, they constitute the largest group of extracellular vesicles. They are involved in intercellular communication mediated by MHC molecules and transport active molecules, including immune regulatory proteins such as PD-L1. ExoPD-L1 can be produced by different cell types, tumor, and non-tumor, and have a greater immunosuppressive effect compared to other forms of sPD-L1 due to their interaction via exosomal MHC I with the TCR, which mimics the effect of the tumor surface PD-L1 [14,15,16,17,18,19,20,21]. Tumor secretion of exoPD-L1 has a direct local impact on various types of tumor microenvironment (such as chronic lymphocytic leukemia (CLL), hepatocellular carcinoma (HCC), glioblastoma (GBM)), where monocytes and macrophages are important components of microenvironment, either through the inhibition of T cells or by promoting PD-L1 upregulation in other immune cells, a major established immune evasion mechanism [22]. When it comes to the lymphatic level, exoPD-L1 will inhibit the production of memory T cells, favoring lymphatic tumor spread. ExoPD-L1 plasma release has also been proved to be responsible for distant immunosuppression [22,23].

In addition to its immunosuppressive role, the most recent evidence places exoPD-L1 as one of the mechanisms of resistance to anti-PD-L1/PD-1 therapy [23]. Preclinical studies and phase I clinical trials have shown an association between the baseline levels of exoPD-L1 and a low response rate to treatment with ICI in NSCLC, metastatic melanoma, and head and neck tumors [21,22,23,24]. Surprisingly, the behavior of exoPD-L1 after ICI therapy, in contrast to baseline exoPD-L1 levels, appears to stratify metastatic melanoma and NSCLC patients into different categories, with the greatest increases in exoPD-L1 levels after treatment found among responders [25,26]. Yang et al. analyzed 33 patients with metastatic NSCLC, showing that a 1.86-fold or greater increase in exoPD-L1 levels after two months of ICI compared to baseline predicted better PFS, OS, and ORR [26]. This finding opens the door to the possibility of exoPD-L1 monitoring during treatment for the early identification of the subgroup of patients who will benefit most from treatment [25,27].

Therefore, despite some controversial results, in general, both PD-L1s and exoPD-L1 expression seem to have a different predictive value in NSCLC depending on whether they are measured at baseline, where high levels are associated with worse prognosis, or after the first weeks of treatment, where an increase is detected in patients with better outcomes. Further research on this matter is needed to support the previous results. Furthermore, different therapeutic strategies are being developed with the aim of eliminating exoPD-L1 (extracorporeal circulation techniques, pharmacological inhibition of secretion, anti-exoPD-L1 antibodies, etc.). This would pave the way to a new promising therapeutic scenario that would enable a change in the natural history of these tumors by reversing a mechanism of primary resistance to immunotherapy [21,23].

## 3. Circulating Non-Coding RNA

Non-coding RNAs (ncRNA) are RNA molecules that do not contain coding-protein sequences (with some exceptions capable of producing small peptides) but are involved in transcriptional regulation. According to their size, there are two major classes: microRNAs (<200 nucleotides), mainly represented by microRNAs (miRNAs), and long ncRNAs (lncRNAs) (>200 nucleotides). In addition to tissue, they can be found free in body fluids or inside extracellular vesicles, such as exosomes, where they are protected from degradation, making exosomal ncRNAs stable biomarkers. ncRNAs are the main regulators of several cellular processes such as proliferation, gene expression, or apoptosis, so their deregulation might lead to carcinogenesis [28,29]. Therefore, the detection of circulating ncRNA in body fluids could serve as a very valuable non-invasive biomarker for patient selection and monitorization. MiRNAs deregulation may be crucial in cancer due to their ability to alter the expression of oncogenes and tumor suppressor genes, being involved in key carcinogenesis processes such as tumor growth, angiogenesis, or immune evasion [30,31].

In this regard, recent studies have focused on the relevance of free circulating and exosome-derived miRNAs in serum and plasma as potential biomarkers for clinical outcomes in NSCLC during ICIs treatment (Table 1). For instance, Fan et al. found that a serum miRNA classifier containing 10 differentially expressed miRNAs (miR-93, -138-5p, -200, -27a, -424, -34a, -28, -106b, -193a-3p, and -181a) exhibited significantly increased expression levels in patients with NSCLC who responded versus non-responders to nivolumab. That classifier showed high accuracy with AUC = 0.975, 95% CI = 0.875–1.108, *p* < 0.0001. By analyzing expression levels throughout the treatment, they observed an increased expression of these 10 miRNAs from pre-treatment to post-treatment in good responders. They have also shown the predictable value in patients’ survival of miRNAs since patients with the 10 high-expressed miRNAs presented with higher PFS (6.25 months vs. 3.21 months (*p* < 0.001; HR, 0.45; 95% CI, 0.25–0.76)) and OS (7.65 months vs. not reached for median OS (*p* < *0*.001; HR, 0.39; 95% CI, 0.15–0.68)) compared to those with 10 low-expressed miRNAs. Moreover, exosomal lncRNAs (*MALAT1, AGAP2-AS1, ATB, TCF7, FOXD2-AS1, HOXA11-AS, PCAF1*, and *BVAR4*) were found to be significantly overexpressed in good responder patients compared with non-responders (*p* < 0.001) [32]. In another study, Halvorsen et al. identified in the serum of 51 NSCLC patients seven circulating miRNAs (miR-215-5p, -411-3p, -493-5p, -494-3p, -495-3p, -548j-5p, and -93-3p) that were associated with increased OS (differentially expressed in patients with OS > 6 months, *p* = 0.005) after nivolumab therapy (sensitivity 71%, specificity of 90%; AUC = 0.814, in the validation cohort) [33]. Boeri et al. also tested the efficacy of an miRNA signature classifier (MSC) composed of 24 miRNAs in combination with PD-L1 as a prognostic marker in a cohort of 140 advanced NSCLC patients treated with ICIs, showing that the patients with intermediate- or low-risk levels in MSC and/or PD-L1 expression ≥ 50% achieved significantly higher ORR (*p* = 0.0024), and both PFS and OS (*p* < 0.0001) [34].

In addition, some studies highlighted the value of the miRNA-320 family as a potential predictor of response in patients treated with ICIs [28,35]. High levels of miR-320b, -320c, and -320d detected in exosomes pre-treatment were associated with Progressive Disease [28]. Costantini et al. also found a down-expression in miRNA-320b and -375 in patients with clinical benefit (i.e., complete response, partial response, or stability lasting ≥6 months after the initiation of nivolumab treatment). Upon analyzing the target genes, miRNA-320b was found to be associated with proliferation genes (*MYC, TUBB1*), and miRNA-375 with immune-related genes (*JAK2, TGF-b2*), the Wnt/b-catenine (*FZD4, FZD8*), and the Hippo pathway (*YAP1*), all known to be involved in ICIs resistance [35].

All of these results support that the dysregulation of free circulating and exosomal ncRNAs may serve as non-invasive biomarkers for predicting the response to immunotherapy in NSCLC patients. However, some limitations must be taken into account, such as the lack of consensus in isolation methods, the heterogeneity of the studies (pre-analytical features, analysis platforms, and statistical approaches), the use of small cohorts of patients, and the need for external validations. These circulating ncRNAs are really promising since they take advantage of being easily analyzed by readily available technologies, such as qPCR rather than NGS approaches, rendering them suitable biomarkers for disease management in clinical practice [36,37].

## 4. Circulating Immune Cells

Immune cells play a critical role in anti-tumor response, especially when it comes to ICI therapies. Consequently, they have been tested for use as circulating predictive biomarkers of ICI response, with a special focus on different immune components, as depicted in the present section.

In a study with 34 lung cancer participants (28 NSCLC and 6 SCLC), Li et al. have recently performed a comprehensive analysis of several immune populations, showing that higher levels of natural killer (NK) cells and a higher CD4+/CD8+ cell ratio predicted longer PFS at ICI treatment baseline. As far as CD4+ T cells were concerned, their levels at baseline correlated significantly with better radiological response (HR = 0.23, *p* = 0.015). In the NSCLC cohort, patients with lower levels of Tregs (regulatory T cells) at baseline, whether with ICI or ICI combined therapy, achieved a better response. Worthy of note, after two cycles of ICI therapy, absolute CD45+, CD3+, and CD4+ T lymphocytes counts were associated with a radiological response as well (*p* = 0.038) [38].

T-cell immunosenescence is a global remodeling of immune functions related to the chronic antigenic stimulation occurring throughout life. It focuses on the phenotypic characteristics of lymphocytes and refers mainly to low proliferative activity. Senescent T cells can be found in young individuals, but they markedly increase during aging, especially in individuals infected with persistent viruses, such as cytomegalovirus [39]. The markers of T-cell immunosenescence have also been correlated with benefitting the treatment (ICI or platinum-based chemotherapy (PCT)) of NSCLC. For instance, the markers of T-cell immunosenescence on peripheral CD8 T cells (measured by the loss of CD28 and presence of CD57 and killer-cell lectin-like receptor (KLRG1) expression) were correlated with worse ORR (*p* = 0.04), PFS [1.8 vs. 6.4 months, *p* = 0.009], and OS [2.8 (95% CI, 2.0-NR) vs. 20.8 (95% CI, 6.0-NR) months, *p* = 0.02] in NSCLC patients (N = 83) treated with ICIs but not in those who underwent PCT (N = 61) [40].

Microparticles (MPs), consisting of submicron vesicles formed by the budding and shedding of the cell membrane during cell activation or apoptosis, as well as the use of immune cell- and platelet-derived microparticles (PMPs), have also been investigated as biomarkers. PMPs comprise the greatest number of microparticles in the circulating blood and are closely related to higher invasiveness, metastasis, and worse prognosis of cancer [41]. In a study with 50 advanced NSCLC patients treated with pembrolizumab or nivolumab (with or without chemotherapy), the total MPs, PMPs, and T-lymphocyte-derived microparticles (T-LyMPs) count after immunotherapy were significantly higher in patients with disease progression than in those reaching ORR to ICIs (*p* < 0.05) [42].

Neutrophils are the dominant type of immune cell in the NSCLC tumor microenvironment (TME), to which they respond and polarize into distinct phenotypes that operate in an anti-tumorigenic or pro-tumorigenic manner [43]. The clinically relevant adverse impact of a neutrophil predominance, both systemically and in the TME of patients with NSCLC, is exacerbated by the negative prognostic value of both a persistent neutrophilia and a high neutrophil-to-lymphocyte ratio (NLR ≥ 5) [44]. In agreement, Russo et al. reported that high NLR was independently related to poorer OS (*p* = 0.001) and PFS (*p* = 0.028) in advanced NSCLC patients (N = 187) treated with nivolumab. Moreover, regarding platelet-to-lymphocyte ratio (PLR), levels below 200 were associated with longer PFS (*p* = 0267) and OS (*p* = 0.05), as well as higher ORR (*p* = 0.04), and disease control rate (DCR) (*p* = 0.001) [45]. Sun et al. showed that high NLR was also correlated with poor pathological response and shorter PFS in patients with resectable NSCLC undergoing neoadjuvant chemotherapy combined or not with ICI (nivolumab, camrelizumab, or tislelizumab) (N = 168) [46]. Ayers et al. found associations between high NLR and poor OS (HR = 1.66; *p* = 0.019) in patients treated with nivolumab, atezolizumab, or pembrolizumab for metastatic NSCLC at any line of therapy. Sustained high NLR after initiation had an even more profound impact on survival than baseline NLR after 2–8 weeks (HR = 3.43; *p* = 4.23 × 10^−8^) and after 8–14 weeks (HR = 3.86; *p* = 1.43 × 10^−6^), regardless of PD-L1 status [47].

Finally, the combination of routine complete blood count tests with different biomarkers has proven to provide meaningful clinical utility to guide treatment decisions for ICI-based therapy in NSCLC patients. For instance, Ayers et al. demonstrated that mild anemia (defined as hemoglobin (HGB) lower than 12 g/dL) had shown a correlation with response to ICI (N = 129, *p* = 0.02), independently of the NLR. In fact, a combined NLR-HGB biomarker can predict ICI response, showing that patients with pre-treatment high NLR and low HBG present worse OS (N= 123, HR = 2.6, *p* = 3.5 × 10^−5^) [47]. Tanaka et al. recently showed that the Lung Immune Prognostic Index (LIPI) could predict resistance to ICI in patients with previously treated advanced NSCLC, finding that in patients with low PD-L1 treated with chemoimmunotherapy (N = 237), LIPI can especially predict both PFS (HR, 2.75; *p* < 0.001) and OS (HR, 2.01; 95% CI, 1.28–3.15; *p* < 0.001) [48]. In addition, the pre-treatment levels of LIPI, combined with N ≥ 4 and high levels of serum Lactate Dehydrogenase (LDH), negatively correlated with the PFS (N = 113, *p* = 0.0119) [49].

Therefore, the use of circulating immune cells as predictive response biomarkers (Table 2) is still lacking more well-designed studies to definitively support its real power. Even though the use of indexes such as PLR, NLR, and LIPI obtained from routine blood analyses open an interesting opportunity to monitor the response of therapy with a direct impact and dependency on the patient’s immune system status.

## 5. Peripheral Blood Cytokine

In an inflamed tumor microenvironment, closely linked to cancer progression, cytokines are key players, carrying messages between cells and promoting the recruitment of immune cells into the tumor microenvironment. In addition, they can induce the expression of immune checkpoint receptors, influencing the expression of PD-L1 and, consequently, the ICI response. In agreement, recent studies have evaluated the predictive value of these soluble mediators in the serum or plasma of cancer patients as biomarkers of immunotherapy response and/or immune-mediated toxicity. In addition to acting as biomarkers of immunotherapy, these proinflammatory and anti-inflammatory cytokines have as well attracted attention as therapeutic targets since they have the ability to both contribute to the antitumor effect and reduce the incidence of adverse events due to their biological functions.

In this regard, Boutsikou et al. measured several cytokines at the same time in 26 NSCLC patients treated with pembrolizumab or nivolumab in monotherapy. They included the blood collected at diagnosis and 3 months after the start of therapy. Specifically, they used a complete flow cytometry panel, including IFN-γ, TNF-α, IL-1β, IL-2, IL-4, IL-5, IL-6, IL-8, IL-10, and IL-12. A correlation was found between cytokine elevation with a better response to immunotherapy, but no correlation was found between cytokine elevation and PD-L1 expression [50].

Other studies focused on specific cytokines and yielded similar results. For instance, Sanmamed et al. demonstrated that an early decrease in serum IL-8 levels was associated with longer OS in 19 NSCLC patients treated with nivolumab or pembrolizumab [51]. Agulló-Ortuño et al. also measured IL8 using ELISA in plasma from 27 NSCLC patients. At baseline, responder and non-responder patients showed no significant differences (*p* = 0.838), although non-responder patients tend to show higher IL-8 levels [52]. In agreement with Sanmamed et al., ICI patients with early decreases or slight increases in plasma IL-8 levels showed a significantly longer OS (HR 7.49, *p* = 0.025) but with no differences in PFS (*p* = 0.215). Using the same approach, they also measured IL-11, but no correlation with PFS or OS was found [52]. Kauffmann-Guerrero et al. studied 29 stage IV NSCLC patients treated with PD1 checkpoint inhibitors in the second line. Serum samples were obtained before treatment and in the first staging, where cytokine concentration was measured using the Human Cytokine-Inflammation Kit. As in the aforementioned studies, they found that patients with high IL-8 showed to have significantly reduced PFS compared with those presenting low levels (median PFS 4.0 vs. 19.71 weeks, *p* = 0.030) [53].

Several studies have focused on IFN-gamma since it induces a lymphocyte-driven immune response, hereby indicating a synergistic effect with ICI treatment. Hirashima et al. reported that a decrease in its levels was associated with early progression. Thus, they suggest that low levels of IFN-gamma before ICI treatment might be useful for the detection of a poor immunological status, although this study was limited to 29 patients [54]. Constatini et al. included a total of 43 patients treated with nivolumab and collected plasma at diagnosis, before the treatment initiation, and at a response assessment (2 months). Opposite to Hirashima, they found that IFN-gamma levels, determined by ELISA, showed no correlation with the ICI response while being effective in predicting toxicities related to immunotherapy [35]. They also found no association with the response when measuring IL-2 levels [35]. Finally, Kauffmann-Guerrero et al. showed that increased levels of IFN-gamma were found in responder patients, becoming highly predictive of a good and durable ICI response (*p* < 0.001), supporting Hirashima’s results [53].

IL-6 is the most extensively studied cytokine. Ozawa et al. measured serum IL-6 and TNF-α using chemiluminescence enzyme immunoassay and enzyme-linked immunosorbent assay, respectively, in 10 NSCLC taken within 7 days before and after the start with ICIs (nivolumab or pembrolizumab). They observed neither differences between pre- and post-initial levels nor an association with the response. Only when CRP levels (a surrogate marker for IL-6) in 21 additional patients were included along with IL-6 a statistically significant association between either high CRP or IL-6 levels or response was found (*p* = 0.07) [55]. Keegan et al. applied Simoa, a new ultrasensitive ELISA single-molecule array, in 47 metastatic NSCLC patients: 33 were treated with pembrolizumab, 10 received nivolumab, and four received other agents. The plasma was collected before the first doses and at on-treatment (range of 17–196 days on treatment). While baseline IL-6 levels did not correlate with PFS, patients with decreasing IL-6 (reducing >40% from pre-treatment to on-treatment timepoints) experienced a higher PFS than those with stable or increasing IL-6 (11 vs. 4–5 months). Intriguingly, there was a trend for fewer KRAS –mutant cancers in the group with decreasing IL-6 levels and better treatment outcomes, pointing towards a possible association between IL-6 changes and tumor mutations [56]. Finally, Kauffmann-Guerrero et al. described that patients with higher IL-6 levels showed significantly reduced PFS compared with patients with lower values (5.14 vs. 38.57 weeks; *p* < 0.001), supporting the previous results. Interestingly, patients with a long-term response also showed significantly lower levels of IL-6 (*p* < 0.001) [53]. Therefore, decreasing IL-6 appears to be predictive of a longer PFS in all the studies, becoming a promising circulating predictive biomarker. Lastly, Kauffmann-Guerrero et al. also measured IL-10, but no association with a response, OS, or PFS was established, whereas, in the case of Tumor Necrosis Factor-alpha (TNF-α), its low levels were associated with a response, although without reaching the statistical significance [53].

All of these studies, in conjunction, highlight the potential of circulating cytokine levels as a clue to the status of the tumor immune microenvironment and the clinical outcome of ICI treatment, especially for IL-6 and INF-gamma levels (Table 3). However, several factors, such as the low number of patients included in the studies, the differences in treatments and methodology, as well as the need for clear cut-offs, restrict their predictive ability. In this regard, Lim et al. stressed the need to take into account other important issues that might directly affect cytokine levels, such as tumor burden, along with other tumor characteristics, including the number of metastases, size of the primary tumor, or the presence of brain metastasis [57]. In addition, their lack of specificity should be specially taken into account, especially for patients with infectious comorbidities, viral disease, vaccination, or other immunomodulatory therapy [56]. Therefore, circulating cytokines represent promising predictive ICI response biomarkers, but further studies are necessary to finally support their potential and favor their translation into the clinic.

## 6. The Role of ctDNA

Circulating free DNA (cfDNA) are small double-strand DNA fragments released into the bloodstream from cells through apoptosis, necrosis, or even active secretion. The majority of cfDNA is usually derived from normal healthy leukocytes and stromal cells. However, in cancer patients, a small fraction of cfDNA can be shed from the tumor itself, termed ctDNA, thus carrying the genetic and epigenetic modifications characteristic of the tumor of origin [58]. Therefore, ctDNA could represent a good biomarker to classify an NSCLC patient as responding to immunotherapy.

Several studies have evaluated the utility of cfDNA levels in the plasma as predictors for clinical benefit in patients treated with ICIs [59,60,61,62,63], supporting its promising potential as a predictive biomarker. Despite the lack of consensus on the methodology to assess cfDNA levels, since some authors have estimated differences in total cfDNA [59,60] while others have evaluated changes in the tumoral fraction (ctDNA) [61,62,63], recent studies have supported that a decrease in cfDNA levels during therapy is associated with better outcomes. For instance, Alama et al. quantified global cfDNA levels by (directed to *hTERT*) and reported that patients with lower global cfDNA values at baseline experienced longer OS than patients with higher cfDNA levels (HR = 2.89, 95% CI = 1.58–5.29, *p* < 0.001) [59]. However, not only at baseline, but longitudinal analyses have also revealed a predictive value for cfDNA changes measured along with ICI treatment. In this way, Ricciuti et al. showed that an early change in the ctDNA allele fraction (AF) in 62 patients was correlated with radiographic responses and long-term clinical outcomes [64]: an AF decrease between the pre-treatment and first on-treatment blood draw was associated with significantly longer median PFS (8.3 vs 3.4 months, HR: 0.29, *p* = 0.0007) and median OS (26.2 vs. 13.2 months, HR: 0.34, *p* = 0.008). Anagnostou et al. also supported the association between cfDNA clearance and response and how ctDNA-based molecular responses can be detected 8.7 weeks earlier than a conventional response assessment (*p* = 0.004) [65]. Passiglia et al. showed that patients with increased global cfDNA levels during treatment (>20% increase at the sixth week) presented significantly lower OS [60]. Accordingly, Goldberg et al. observed that a decrease of >50% in ctDNA levels, determined by NGS, was significantly associated with superior OS (HR: 0.17; 95%, *p* = 0.007) [62]. Giroux Leprieur et al. also found a significant correlation between ctDNA concentration at the time of the first radiological evaluation and patients’ response, with significantly lower ctDNA levels found in patients with clinical benefit and longer OS and PFS (*p* < 0.0001) [63]. Finally, Guibert et al. also demonstrated that a decrease in ctDNA allelic fraction after 1 month of treatment was related to longer PFS in anti-PD-1 treated NSCLC patients [61].

Moreover, ctDNA can be used as a non-invasive tool for the detection of point mutations associated with sensitivity to immunotherapy. Several studies have correlated point mutations in *STK11* with a lack of benefit in NSCLC patients treated with ICIs [61,66,67,68]. Guibert et al. evidenced that the presence of mutated *PTEN* or *STK11* was correlated with poor outcomes (HR: 8.9, *p* = 0.09 for *PTEN*; HR: 4.7, *p* = 0.003 for *STK11*) [61]. In contrast, they found that transversion mutations (changes between purine and pyrimidine bases) in *KRAS* and *TP53* genes could predict a better response (HR: 0.36, *p* = 0.011 for *TP53*; HR: 0.46, *p* = 0.11 for *KRAS*). Accordingly, they proposed an algorithm to classify patients with high or low “immune scores” to predict those patients more likely to benefit from ICIs therapy. Thus, they define “High immune score” as those patients with no targetable driver mutations (*EGFR, ROS1, ALK,* and *BRAF V600E*), no mutations in *PTEN* or *STK11* but harboring transversion mutations in *TP53* and *KRAS* [61].

Intriguingly, *STK11* mutations are often associated with *KRAS* mutation [67], and it has been reported that *STK11/KRAS* co-mutations cause worse survival outcomes in ICI-treated patients [69]. This co-mutation has been correlated to the suppression of the tumor immune surveillance response and, therefore, a low-activity tumor microenvironment [70]. In fact, a recently published study reported a negative impact of *KRAS/STK11* (HR = 10.936; 95%, CI: 2.337–51.164, *p* = 0.002) and *KRAS/STK11/TP53* (HR = 17.609; 95%, CI: 3.777–82.089, *p* = 0.01) co-mutations in ICI-treated patients, both in OS and PFS, supporting the potential role of this co-mutation as a predictive biomarker to be tested in plasma by NGS identification [68]. In general, *STK11* mutations have been strongly associated with non-responder NSCLC patients. Consequently, the analyses of *STK11* mutations in cfDNA could help to detect likely non-responder patients. Another frequent co-mutation in *KRAS*-mutant NSCLC patients was found in *KEAP1/NFE2L2*, which has been proposed as a predictive factor of shorter OS in patients receiving ICIs in monotherapy [67]. In agreement, Zhu et al. also observed a decrease in OS in *KEAP1/NFE2L2*-mutant patients. Nevertheless, no significant differences in OS were reported among the patients treated with immunotherapy and chemotherapy, suggesting that *KEAP1/NFE2L2* mutations are associated with a worse prognosis for both types of treatment [71]. Of note, in a recent work by Ricciuti et al., *KEAP1* and *STK11* mutations were shown to confer worse outcomes (both shorter PFS and OS) only in *KRAS*-mutated lung adenocarcinoma patients but not among patients with *KRAS* wild-type [72]. Although the molecular mechanisms behind these correlations must be further explored, these results highlight the importance of considering *KRAS* mutation status to stratify patients receiving ICIs. *KRAS* mutation has also been tracked in plasma by droplet digital-PCR (ddPCR), which easily allows the monitorization of changes in the abundance of mutated alleles along the course of the treatment. In fact, an increase in the mutated fraction has been correlated with a worse prognosis, especially in those patients who are negative at baseline and become positive after 3 or 4 weeks (shorter PFS and OS) [73].

In addition, two acquired mutations in β2-microglobulin (*B2M*), a component of MHC class-I, have also been identified in ctDNA as resistant to ICIs [74].

Finally, in addition to being a candidate biomarker to predict ICI resistance, ctDNA can also be tracked to monitor mutations conferring better outcomes, such as *ARID1A* and *ARID1B*, which have been associated with better response to the treatment and longer PFS [75].

Overall, these results support the utility of cfDNA/ctDNA biomarkers in liquid biopsy for the prediction of the ICI response (Table 4). ctDNA analysis has become a promising tool useful for patient selection in clinical practice, but also for monitoring along the course of the treatment. Moreover, it can be easily analyzed by target approaches directed to a few relevant genes, such as ddPCR and qPCR [76,77], or by NGS larger gene panels, which are available in commercial kits, including the analysis software [78]. However, it remains important to be aware of the limitations of these studies, as occasionally, the number of patients is limited, and there is a lack of validation cohorts. Particularly, patient cohorts are frequently heterogeneous in the treatment regimen, with different drugs and lines of treatment. Nevertheless, ctDNA analysis shows very promising results, and continuing in this line of research is essential to improve our knowledge of predictive biomarkers for ICIs response and facilitate its applicability in clinical settings.

## 7. Circulating TMB (bTMB)

The blood Tumor Mutational Burden (bTMB) or circulating TMB is defined as the total number of somatic mutations detected on a tumor genome from ctDNA, in contrast to the classic tissue TMB assessed in tissue biopsy samples (tTMB) [79,80]. As discussed in the introduction, high TMB has been widely associated with benefits from ICIs in NSCLC patients [81,82,83,84,85], but important drawbacks limit its translation into the clinic: firstly, TMB was originally based on Whole-Exome Sequencing (WES), which entails the need for well-conserved genomic concentrated DNA, with a subsequently high sequencing cost, and the need for extensive bioinformatic analyses [79,84,85,86]. Secondly, there is a lack of consensus over the cut-off points for stratifying patients based on their TMB levels as well as on the types of mutations that should be considered [84,85]. Therefore, studies have been shifting towards the estimation of targeted gene panels in a search for faster and more affordable ways to assess tTMB [79,86]. However, the lack of clinical and analytical validation of the existing panels, coupled with the need for standardization, obscured their clinical translation [79,81,86,87]. Figure 2 illustrates the lack of standardization for TMB assessment methodologies, criteria, and thresholds.

Based on this scenario, bTMB emerged to overcome the current limitations of solid tumor biopsies, as it allows a more realistic vision of the tumor characteristics and progression (Figure 2). Gandara et al. demonstrated for the first time that bTMB could be measured accurately and reproducibly in plasma [80], using a panel of 395 cancer-related genes, including all somatic SNVs with AF (Allele Frequency) ≥0.5%. This method was validated using data from two randomized clinical trials: the POPLAR (NCT01903993) study (N = 284) [88], which compared second-line/third-line ICI atezolizumab with the standard of care (docetaxel), and the OAK trial (NCT02008227) (N = 1225) [89], a randomized phase III study that compared atezolizumab with docetaxel in metastatic NSCLC. The results showed an association between bTMB and better PFS and OS with ICIs versus chemotherapy-treated patients [80]. Notably, when they performed a comparison between bTMB and tTMB, the analyses a showed positive correlation, although no identical variants were found.

In line with these findings, several studies have recently assessed the potential of bTMB as a predictive ICI biomarker in NSCLC. The MYSTIC clinical trial (NCT02453282) (N = 1118) [90] was the first phase 3 clinical trial using bTMB as a selection tool. Interestingly, bTMB was assessed in 72.4% of the patients compared to the 41.1% of participants in which tTMB was assessed. The results supported that plasma evaluation represented a more readily available source of the tumor genome. In terms of outcomes, bTMB was found to be predictive of increased OS (HR,0.49; 95% CI, 0.32–0.74), better PFS, and ORR in ICI-treated patients.

Further studies assessing the predictive accuracy of circulating TMB again revealed differences between bTMB and tTMB [91,92]. Zhang et al. suggested that, despite the differences between bTMB-tTMB, bTMB could still be predictive of outcomes, underscoring the importance of the quality of included neoantigens rather than the total amount of mutations [92]. This is in line with the proposed idea of shifting towards the identification of specific mutations in cfDNA that elicit the generation of highly immunogenic peptides [93]. As an example, Wei et al. performed a meta-analysis of the published literature to evaluate the prognostic value of ctDNA and bTMB in patients receiving ICIs [94]. Their results showed that, unlike ctDNA clearance, bTMB could not be used independently as a prognostic factor for the response of patients undergoing ICI due to the current limitations of detection technology and a lack of standardization.

In the same line, other studies proposed the combination of bTMB with other parameters to improve its predictive capacity [91,92,93,94,95]. For instance, a retrospective analysis of POPLAR and OAK studies again supported that bTMB alone may be insufficient to predict ICI outcomes and advised to use it in combination with Maximum Somatic Allele Frequency (MSAF): a parameter that helps to estimate the fraction of tumor cfDNA in peripheral blood samples [95]. Consequently, MSAF may partially explain the differences found between blood and tissue TMB since a low MSAF could lead to a lower detection of tumor somatic mutations in plasma, causing patients with low MSAF and tTMB-high to be misclassified as bTMB-low. In agreement, MSAF in combination with bTMB has been suggested to effectively predict the OS and PFS benefits of ICI vs. chemotherapy [95]. An additional proposed solution was the inclusion of parameters that assess tumor heterogeneity [91,92,93,94,95,96]. In a letter to the editor on the Wang et al. study, Liu et al. stated that bTMB is likely to carry two sorts of bias: the first one is related to tumor size: larger tumors shed more cells, leading to a higher presence of ctDNA in the bloodstream. The second is tumor heterogeneity, as low-frequency mutations are more likely to be minor subclonal [96,97]. When Liu et al. recalculated the bTMB after adjusting for both factors, the correlation of bTMB with PFS, OS, and OOR was enhanced as compared to the original results. With these claims in mind, Fridland et al. proposed an additional index called Tumor Heterogeneity Index (THI), which may well determine whether a given mutation is subclonal, and calculate a score reflecting tumor heterogeneity, consequently giving a less biased vision of tumor mutational landscape than tTMB or bTMB alone [91].

In summary, bTMB is the most studied circulating predictive biomarker; thus, it has the most well-supported results, even based on international clinical trials. Its advantages over tTMB seem to be evident, and several studies support its capacity to predict the response, along with or in combination with other markers. Therefore, nowadays, it still represents the more reliable biomarker to predict response using liquid biopsy. The main disadvantage of this biomarker relies on the technology required for its estimation. Therefore, unless more cost-effective and standardized approaches for TMB estimation are finally implemented, it might be replaced in the future by other candidates, such as those other than the here presented biomarkers, which are based on more affordable technologies.

## 8. Conclusions and Future Directions

Liquid biopsy is taking an increasingly crucial role in cancer management as a non-invasive source of biological samples, not only from the tumor tissue but also from the tumor microenvironment, which may serve as valuable biomarkers for diagnosis, prognosis, and the prediction of treatment response. This technique for sample collection takes special relevance in the case of NSCLC, the leading cause of cancer-related deaths, since tissue biopsies are usually small specimens (with high intratumor heterogeneity bias), difficult to obtain, and highly invasive for the patient. Unlike tissue biopsy, liquid biopsies are easy to obtain, reflect the overall tumor state, represent better tumor heterogeneity, and allow for the tracking of the tumor evolution along the course of the treatment in real-time (Table 5).

Nowadays, the measure of PD-L1 expression levels in tissue is currently approved for patient selection. Therefore, the possibility of dtermining expression levels in blood samples represents a gateway to predicting ICI response in a non-invasive manner during the treatment, with the possibility of evaluating the evolution of PD-L1 levels over time. The previous results support that sPD-L1 levels can be used to predict ICI response, PFS, and OS in advanced solid tumors, while the value of sPD-L1 changes still needs validation. In addition, exo-PD-L1 levels have been evaluated before and during treatment, and although the results are still inconsistent, a future perspective of treatment and monitorization has been opened.

The dysregulation of circulating ncRNAs has also demonstrated its value as a non-invasive biomarker for predicting the response to ICIs in NSCLC patients, based on several miRNAs and lncRNAs identities and signatures (Table 1). Their evaluation by qPCR or NGS, together with the possibility to isolate them from exosomes, where it is protected from rapid degradation, makes them a suitable biomarker to include in clinical practice. However, a future validation in independent cohorts is required since there is high heterogeneity in the studies regarding the patient’s pre-clinical features, RNA isolation methods, and the expression analysis that should be resolved towards its translation into clinics.

The state of TME is a well-known major player in the response of ICIs, and therefore the counting of different circulating immune cell populations, such as T-cells, MPs, or neutrophils, has also been recently correlated with immunotherapy response. In this sense, although more studies and studied patients are desirable, combined biomarkers based on routine blood count tests, such as HGB levels, have been proposed to guide treatment decisions with promising results.

In addition to immune cells, cytokines secreted to the tumor environment can also inform clinicians about the inflammatory state of the tumor and the surrounding TME, and its correlation with the efficacy of immunotherapy has also been studied with remarkable results for both IL-6 and INF-gamma levels.

The discovery of somatic mutations in genes associated with sensitivity or resistance to immunotherapy in cfDNA will improve clinical decision-making regarding treatment regimens, with the possibility of using NGS panels with a reduced number of genes or high-sensitivity methods, such as ddPCR, obtaining clear and easy to interpret results, allowing for the easy translation to clinical practice. In addition, the global levels of ctDNA in the plasma have been proposed as an indicator of treatment response since a decrease in the ctDNA is associated with better outcomes in ICI-treated patients. However, there is not a defined consensus on the methods for measuring the overall levels. As an alternative, evaluating the bTMB has been suggested as an alternative to tTMB. Consequently, a higher bTMB has been associated with clinical benefits from ICIs. Nevertheless, determining the bTMB still requires high-quality DNA, high sequencing costs, and the difficulty of data analysis.

Finally, different cofounder factors are known to have a role in the ICIs response, and they would need to be considered when evaluating candidate biomarkers with a predictive response value. Deshpande et al. have recently reviewed how lifestyle habits (i.e., exercise and diet) favor responsiveness while alcohol consumption mitigated the effect of ICIs [98]. Diet has also been recognized to impact the response: obesity, for instance, enhances PD-1 expression, while an appropriate intestinal microbiota might modulate the immune system, affecting ICI response [98,99,100]. Differences in ICI response between current/former smokers and never-smokers have also been recently reported [101]. In addition, blood-based biomarkers also present preanalytical possible confounding factors that should be carefully evaluated and considered since their levels can be affected by daily variations linked to circadian rhythm, digestion, physical activities, co-infections, or treatment regimens. Considering these factors and reaching a consensus on the best technical procedure in the ongoing studies would improve the reliability of liquid-biopsy-based monitoring.

To sum up, in spite of the need for larger study cohorts and independent validations to reach a consensus that eases its translation into clinical practice, the potential value of blood-based biomarkers to predict ICI response in NSCLC patients is already unquestionable, as highlighted by all the studies included within this review. They evince their potential to not only monitor the tumor status but also the TME along the course of therapy in a non-invasive way, opening up promising approaches that will shape the future of immunoncology, and the future seems to be headed towards their implementation.

## Figures and Tables

**Figure 1 cancers-14-02626-f001:**
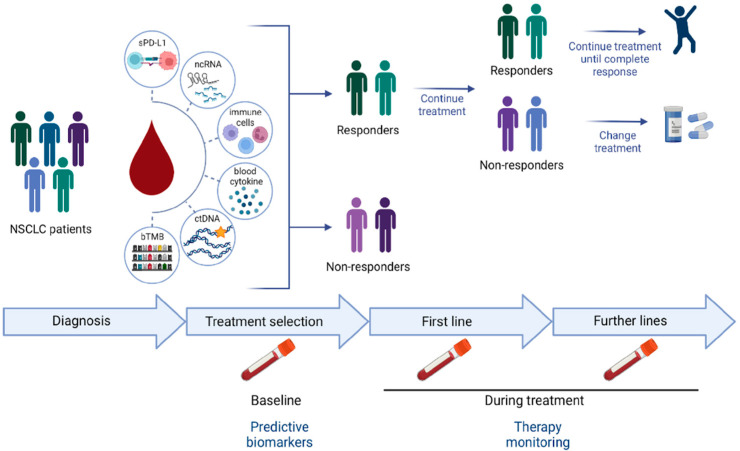
Potential clinical applications of liquid biopsy: soluble biomarkers can be used for ICI response prediction at baseline prior to treatment selection, enabling tracking of tumor evolution during the treatment.

**Figure 2 cancers-14-02626-f002:**
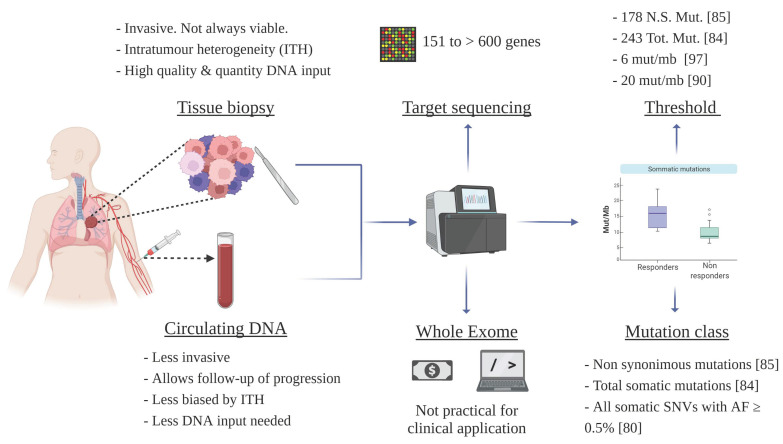
Lack of standardization for TMB assessment methodologies, criteria, and thresholds. Here we represent the overview of the subsequent stages of TMB determination, which still lacks standardization. N.S. Mut.: Nonsynonymous Mutations; Tot. Mut: Total Mutations; mut/mb: Mutations per megabase; SNVs: Single nucleotide variations.

**Table 1 cancers-14-02626-t001:** Circulating ncRNA identities identified as potential biomarkers for ICI response prediction in NSCLC.

Study	Cohort	Treatment	Method	Source	Selected Biomarkers
[32]	9 NSCLC (stage III-IV)	Anti-PD-1, anti-PD-L1	NGS	Exosomal miRNA	miR-320b, -320c, -320d
[33]	80 NSCLC (stage IV)	Anti-PD-1 (nivolumab)	qPCR	Serum miRNA	miR-93, -138- 5p, -200, -27a, -424, -34a, -28, -106b, -193a-3p, and -181a
				Exosomal lncRNA	*MALAT1, AGAP2-AS1, ATB, TCF7, FOXD2-AS1, HOXA11-AS, PCAF1*, *BVAR4*
[34]	51 advanced NSCLC	Anti-PD-1 (nivolumab)	NGS/qPCR	Serum miRNA	miR- 215-5p, -411-3p, -493-5p, -494-3p, -495-3p, -548j-5p, -93-3p
[28]	140 NSCLC (stage III-IV)	Several ^1^	qPCR	Plasma miRNA	miR-101-3p, -106a-5p, -126-5p, -133a, -140-3p, -140-5p, -142- 3p, -145-5p, -148a-3p, -15b-5p, -16-5p, -17-5p, -197-3p, -19b-3p, -21-5p, -221-3p, -28-3p, -30b-5p, -30c-5p, -320a, -451a, -486-5p, -660-5p, -92a-3p
[32]	18 advanced NSCLC	Anti-PD-1 (nivolumab)	NGS	Plasma miRNA	miR-320b, -375

^1^ Anti-PD-1 (nivolumab and pembrolizumab), anti-PD-L1 (avelumab, atezolizumab, durvalumab) and combined anti-PD-1 and anti-CTLA4 (durvalumab + tremelimumab).

**Table 2 cancers-14-02626-t002:** Circulating immune cells studies assessing different immune cell-related biomarkers for ICI response prediction and their association with clinical outcomes.

Biomarker	References	Outcomes
Presence of NK cells & CD4+/CD8+ ratio	[38]	Longer PFS, better response to ICIs at baseline
T-cell immunosenescence	[40]	Worse ORR, PFS and OS
Microparticles (PMPs)	[41,42]	High levels associated with worse prognosis
Neutrophil-to-lymphocyte ratio & platelet-to-lymphocyte-ratio	[44,45,46,47]	Higher levels correlate with shorter OS, PFS, worse ORR and poor response
LIPI	[48,49]	Resistance to ICI, negative correlation with PFS

**Table 3 cancers-14-02626-t003:** Peripheral blood cytokines most studied and predictive values relative to ICI response.

Biomarker	References	Outcomes
IL-8	[51,52,53]	Early decreases associated with better prognosis
IFN-gamma	[35,53,54]	Increased levels predictive of a good response, or association with toxicities
IL-6	[53,55,56]	Early decreases associated with better prognosis or no association with response

**Table 4 cancers-14-02626-t004:** Circulating free DNA studies evaluating the potential of its levels or mutated individual genes as predictors of ICI response and the association with patients’ outcomes.

Biomarker	References	Outcomes
cfDNA levels at baseline	[59]	Low levels are associated with higher OS
cfDNA levels during treatment	[60,61,62,63]	Decrease global levels related to better outcomes
*STK11*	[61,68]	Mutations associated with worst outcomes
*PTEN*	[61]	Mutations associated with worst outcomes
*KRAS*	[61,68,69,70,72]	Transversions related with better outcomes Co-mutations with other genes associated with resistance
*TP53*	[61,68]	Transversions related with better outcomes
*KEAP1/NFE2L2*	[67,71]	Shorter OS
*B2M*	[74]	Resistance to ICIs
*ARID1A, ARID1V*	[75]	Better response and longer PFS

**Table 5 cancers-14-02626-t005:** Summary of advantages and limitations of reviewed liquid biopsy biomarkers.

Biological Source	Methods for Detection	Importance	Limitations
Soluble and exosomal PD-L1 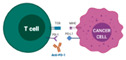	ELISA, isolation of exosomes	- PD-L1 assays established as drug companion diagnostics - Non-invasive and unbiased alternative to tissue PD-L1 determination	- Studies with a limited number of patients - Lack of results validation in more patients and independent cohorts
Circ ncRNA 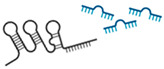	NGS (target panels), qPCR	- Low amount of starting material is required - High expression stability - Easy to evaluate (qPCR) - Available supported ncRNA identities and signatures to predict response	- Lack of standardization in isolation methods - High heterogeneity among results and study design
Circulating immune cells 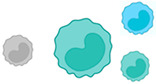	Flow cytometry	- Representation of the TME - Easy analysis technology (flow cytometry and routine blood analysis)	- Studies with a limited number of patients - High heterogeneity among the variables evaluated
Peripheral blood cytokine 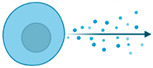	Flow cytometry panels, ELISA	- Informative of the inflammatory state of the tumor and TME - Easy to evaluate - Available supported candidates	- Studies with a limited number of patients - Differences in treatments and methodology - Not established cut-offs - Highly variable levels depending on current patient state
ctDNA 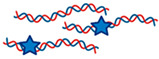	NGS (gene panels), qPCR, ddPCR	- Low amount of starting material is required - Well-established methodology - Easy evaluation and result interpretation - Allows tumor mutations monitorization during treatment - Available supported gene candidates to predict response	- Large gene panels are difficult to analyze bioinformatically - Heterogeneous cohorts (different treatments and lines of therapy) - Lack of validation cohorts
bTMB 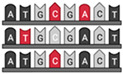	WES, Targeted gene panels	- Reflects the current tumor state in a non-invasive and unbiased alternative to tTMB determination - Available supported candidates and signatures to predict response	- Needs high-quality DNA - Expensive (sequencing costs) and difficult to analyze - Lack of consensus on the cut-off points and genes considered in its calculation - Lack of clinical and analytical validation of gene panels - Hard to translate to clinical practice

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
