# Peer review of "Emerging Blood-Based Biomarkers for Predicting Immunotherapy Response in NSCLC"

_cancers, 2022, doi:10.3390/cancers14112626_

Round 1

Reviewer 1 Report

In present review, authors discuss the implications of non invasive  blood based biomarkers for immunotherapy efficacy. I have several reservations, my comments are appended as below:

  1. Authors should first note statistics on immunotherapy efficacy and resistance.
  2. Are there any longitudinal studies focusing on blood based markers as shown in figure 1? This would be a valuable information.

3.Along with the other cofounders, there sre factors such as obesity, smoking known to impact immunotherapy efficacy. Author’s should discuss PMID: 33076303 and note if there are studies conducted in patients with smoking history or obesity. It could be translationally relevant information.  

  1. While quoting studies with patients, annotate with details such as number of patients and statistical inference (HR, P value). For instance, reference 8,9, line 115.
  2. Authors quote a good account of sPDL1. Are there studies noting its release in blood depending on pathological grade of tumor or in patients previously treated with immunotherapy and as marker of recurrence?
  3. Line 95, reference 17, line 216, reference 20- specify the pathologies. This should be followed in complete manuscript.
  4. Each section should end with conclusion and future directions in brief.
  5. Circulating immune cells and cytokine- present a table. Is there any record of Treg cells or macrophages?
  6. Author’s should highlight clinical studies using markers listed in table 1. Are there any commercial kits detecting the same?
  7. There should be ‘future directions’ section.

Reviewer 2 Report

A fairly informative review of blood-based biomarkers for predicting immunotherapy response in NSCLC is presented for review. I have a few comments for the authors:

  1. I would like to see the methodology for preparing the review, on what basis the selection of publications for preparing the review was carried out, what keywords were used for the search, what databases, etc.
  2. By analogy with Table 1, it would be clearer to provide summary tables for cytokines, circulating DNA, etc.
  3. Figure 3 is poorly perceived in the form of a figure, it is better to perform it in the form of a table.

Reviewer 3 Report

The authors provide a comprehensive review of studies evaluating liquid biopsy biomarkers as clinical tools to measure response to immunotherapy in NSCLC. The authors have accurately reported the findings in an unbiased manner. The figures and tables have been thoughtfully constructed to summarize the results for the readers.

Minor suggestion- one of the other factors for consideration when using liquid biospy is the timing of blood draw, as several factors such as time of day(circadian rhythm)  and even treatment regimens can influence the blood-based biomarkers. For example, chemotherapy treatment can result in increased circulating tumor DNA, cytokines etc in the peripheral blood which could be misleading. They authors may consider adding these confounding factors if not already included. Otherwise this is a very well written review.

Round 2

Reviewer 1 Report

I congratulate the authors for providing modifications, with this, the manuscript is in improved form. I however suggest taking care of minor concerns:

1.I observe that the first author name in suggested reference 100 is not matching. It should be corrected. 

Reviewer 2 Report

The authors have substantially revised the manuscript; in its present form, I recommend accepting it for publication.
